# Carbon Capture Using Porous Silica Materials

**DOI:** 10.3390/nano13142050

**Published:** 2023-07-11

**Authors:** Sumedha M. Amaraweera, Chamila A. Gunathilake, Oneesha H. P. Gunawardene, Rohan S. Dassanayake, Eun-Bum Cho, Yanhai Du

**Affiliations:** 1Department of Manufacturing and Industrial Engineering, Faculty of Engineering, University of Peradeniya, Peradeniya 20400, Sri Lanka; sumedha.sma@gmail.com; 2Department of Chemical and Process Engineering, Faculty of Engineering, University of Peradeniya, Peradeniya 20400, Sri Lanka; hishendri1995@gmail.com; 3Department of Applied Engineering & Technology, College of Aeronautics and Engineering, Kent State University, Kent, OH 44242, USA; 4Department of Biosystems Technology, Faculty of Technology, University of Sri Jayewardenepura, Homagama 10200, Sri Lanka; 5Department of Fine Chemistry, Seoul National University of Science and Technology, Seoul 01811, Republic of Korea

**Keywords:** CO_2_ capture technologies, CO_2_ adsorption, porous silica, amine functionalized porous silica, decarbonization

## Abstract

As the primary greenhouse gas, CO_2_ emission has noticeably increased over the past decades resulting in global warming and climate change. Surprisingly, anthropogenic activities have increased atmospheric CO_2_ by 50% in less than 200 years, causing more frequent and severe rainfall, snowstorms, flash floods, droughts, heat waves, and rising sea levels in recent times. Hence, reducing the excess CO_2_ in the atmosphere is imperative to keep the global average temperature rise below 2 °C. Among many CO_2_ mitigation approaches, CO_2_ capture using porous materials is considered one of the most promising technologies. Porous solid materials such as carbons, silica, zeolites, hollow fibers, and alumina have been widely investigated in CO_2_ capture technologies. Interestingly, porous silica-based materials have recently emerged as excellent candidates for CO_2_ capture technologies due to their unique properties, including high surface area, pore volume, easy surface functionalization, excellent thermal, and mechanical stability, and low cost. Therefore, this review comprehensively covers major CO_2_ capture processes and their pros and cons, selecting a suitable sorbent, use of liquid amines, and highlights the recent progress of various porous silica materials, including amine-functionalized silica, their reaction mechanisms and synthesis processes. Moreover, CO_2_ adsorption capacities, gas selectivity, reusability, current challenges, and future directions of porous silica materials have also been discussed.

## 1. Introduction

With the exponential growth of industrialization, global warming and climate change have become worldwide concerns and have attracted much attention in recent decades [1]. Furthermore, human activities have significantly contributed to the increased levels of CO_2_ in the atmosphere. For example, atmospheric CO_2_ measured at NOAA’s Mauna Loa Atmospheric Baseline Observatory peaked for 2021 at a monthly average of 419 parts per million (ppm), and it is reported as the highest level since accurate measurements began 63 years ago [2]. 

The increase in CO_2_ concentration leads to the rise in global temperature and sea levels, alternative of rainfall patterns, extinction of species, natural disasters such as severe weather events, ranging from flash floods, hurricanes, freezing winters, severe droughts, heat waves, urban smog, and cold streaks [3]. 

The main CO_2_ stationary emission sources are power plants, refineries, chemical and petrochemical, iron and steel, gas processing, and cement industries. More irreversible and adverse environmental impacts should be expected if atmospheric carbon dioxide continues to rise. Therefore, the international communities led by the United Nations reached a landmark global accord, the Paris Agreement, adopted by 196 nations in 2015 to address climate change and related issues. Moreover, countries around the globe made their “nationally determined contributions (NDCs)” of greenhouse gas reduction. Different approaches employed in different countries to reduce CO_2_ emissions are shown in Table 1. Table 1 also summarizes the major advantages and disadvantages of each approach.

Among these approaches, the CO_2_ capture and storage (CSS) can reduce CO_2_ emissions by 85–90% from large emission sources [4]. CCS includes different CO_2_ capture, separation, transport, storage technologies, and chemical conversion, which are discussed in detail below.

## 2. CO_2_ Capture

### 2.1. CO_2_ Capture Technologies

Capture and sequestration of CO_2_ (CCS) from aforementioned stationary emission sources has been identified as a paramount option for the issues of global warming and climate change. CCS includes four primary steps known as CO_2_ capture, compression, transport, and storage, therefore, developing an efficient and economically feasible technology for the capture and sequestration of CO_2_ produced by anthropogenic emissions is critically important. CO_2_ capture is the central part of the CCS technology process and gained around 70–80% of the total expensive. However, CSS methods can be classified as, for example, (i) Post-combustion (ii) Pre-combustion, and (iii) Oxy-fuel combustion (Oxygen-fired combustion) [5,6].

In *post-combustion capture* technology, it collects and separates the CO_2_ from the emission gases of a combustion system [7,8,9,10,11]. Firstly, flue gas (mainly consists of CO_2_, H_2_O, and N_2_) passes through denitrification and desulphurization treatments. As the next step, the flue gas is fed to an absorber which contains solvent. Herein, CO_2_ regeneration occurs. Then the CO_2_-rich absorbent is sent to a CO_2_-stripper unit to release the CO_2_ gas. Moreover, CO_2_-lean absorbent is sent back to the CO_2_-absorber unit [1]. Next, the captured CO_2_ is then compressed into supercritical fluid and then transported [1] as shown in Figure 1.

*Pre-combustion capture* is a technology where CO_2_ is captured before the combustion process and CO_2_ is generated as an intermediate co-product of conversion process [12]. The pre-combustion technologies are mainly used in power plants, production of fertilizers and natural gas [13,14]. 

In *oxyfuel combustion*, the carbon-based fuel consumes in re-circulated flue gas and oxygen (O_2_) stream. CSS capture technology is considered expensive due to the high cost of O_2_ separation and production. However, the capture and separation of CO_2_ are reasonably easy compared to other methods and is considered as an energy-saving method [15].

Among the currently available technologies, post-combustion capture has grabbed much attention because it can be easily accomplished, applicable for large scale-power plants, easily managed and required short time for CO_2_ capture compared to other available methods [1]. Post-combustion capture uses different methods for gas separation, and collects CO_2_ by adsorption/desorption, as shown in Table 2, including absorption [6,16], adsorption [6,17], membrane-based technologies [18,19], and cryogenics [20]. Table 2 also depicts the efficiency, advantages, and disadvantages of the different types of post-combustion capture technologies.

Absorption process mainly uses liquids to capture CO_2_. During adsorption, once CO_2_ is separated from the gas, the sorbent should be regenerated by using a stripper, heating, or depressurization. Moreover, this method is considered as the most established process for CO_2_ separation [21]. In general, adsorbents can be divided into two types, namely, chemical and physical adsorbents (see Table 2 for details).

### 2.2. Criteria for Selecting CO_2_ Sorbent Material

Certain economical and technical properties are required in order to select the best solid adsorbent candidate for a particular CO_2_ capture application. These criteria are listed and described below.

Adsorption capacity for CO_2_:

The equilibrium adsorption capacity of a sorbent material is represented by its equilibrium adsorption isotherm. The adsorption capacity is an important parameter when considering the cost. Moreover, which causes reduction in the sorbent quantity, and in the size of the adsorption column. However, to enhance the adsorption capacity of solid sorbents, functionalization has been carried out with existing monoethanolamine (MEA) [24]. The CO_2_ working capacity should be in the range of 2–4 mmol/g of the sorbent [25].

Selectivity for CO_2_:

The adsorption selectivity or selectivity of CO_2_ is explained as the sorption uptake ratio of a target gas species compared to another type (as example N_2_) contained in a gaseous mixture under given operation conditions. Therefore, it depends on the purity of the adsorbed gas in the effluent [21]. However, the purity of CO_2_ influences transportation and sequestration and, therefore, this criterion plays an important role in CO_2_ sequestration [24].

Adsorption and desorption kinetics:

It is necessary to have fast adsorption/desorption kinetics for CO_2_ and it controls the cycle time of a fixed-bed adsorption system. Fast kinetics results in a sharp CO_2_ breakthrough curve in which effluent CO_2_ concentration changes are measured as a function of time, while slow kinetics provides a distended breakthrough curve. However, both fast and slow adsorption and desorption kinetics impact on the amount of sorbent required. In functionalized solid sorbents, the overall kinetics of CO_2_ adsorption mainly depend on the functional groups present, as well as the mass transfer or diffusional resistance of the gas phase through the sorbent structures. The porous support structures of functionalized solid sorbents also can be tailored to minimize the diffusional resistance. The faster an adsorbent can adsorb CO_2_ and be desorbed, the less of it will be needed to capture a given volume of flue gas [24].

Mechanical strength of sorbent particles:

The sorbent must show the stable microstructure and morphological structure in adsorption and regeneration steps. Mainly disintegration of the sorbent particles occurs due to the high volumetric flow rate of flue gas, vibration, and temperature. Apart from that, this could also happen due to abrasion or crushing. Therefore, a sufficient mechanical strength of a sorbent particles is required to keep CO_2_ capture process cost-effective [24].

Chemical stability/tolerance towards impurities:

Solid CO_2_ capture sorbents such as amine-functionalized sorbents should be stable in an oxidizing environment of flue gas and should be resistant to common flue gas contaminants [24].

Regeneration of sorbents:

The regeneration of the sorbent is energy saving and is one of the most important parameters required for improving energy efficiency [26]. Regeneration can be achieved through the adjustment of the thermodynamics of the interaction between CO_2_ and the solid adsorbent [24]. Considering regeneration, physisorption is mostly favored over chemisorption since the latter involves high energy consumption for regeneration.

Sorbent costs:

The production cost is the main key point when considering industrial applications at reasonable gas selectivity and adsorption performance [24].

### 2.3. Liquid Amine for CO_2_ Capture

Development of solvents for CO_2_ chemical absorption is a major area of research [27]. The ideal solvent should have a high CO_2_ absorption capacity and react rapidly and reversibly with CO_2_ with minimal heat requirement. The solvent should exhibit the following properties such as stability in oxidative and thermal environment, low vapor pressure, toxicity, flammability, and reasonable production cost [27].

Recently, a most promising CO_2_ capture method with chemical absorption is by using liquid amine which can be divided mainly into two groups known as simple alkanolamines and sterically hindered amines [28]. Examples for simple alkanolamines are monoethanolamine (MEA), diethanolamine (DEA), and triethanolamine (TEA) [29,30]. Furthermore, alkanolamines are the most widely used sorbents for CO_2_ capture. The structures of alkanolamines include primary, secondary, ternary amines containing at least one hydroxyl (-OH) group and amine group-(N-R) as shown in Table 3.

However, these different amine classes have different reaction kinetics with CO_2_, CO_2_ absorption capacity and equilibria, stability, and corrosion [28]. Advantages and disadvantages among the alkanolamines are shown in Table 3. As shown in Equations (1) and (2) below, both primary and secondary amines react with CO_2_ to form a carbamate and protonated amine, consuming approximately two moles of amine per mole of CO_2_ according to the zwitterion mechanism [31]. According to Equation (3), tertiary amines react with CO_2_ gas molecules in the presence of H_2_O while forming bicarbonates.
(1)$${CO}_{2}+2R_{1}{NH}_{2}\overset{}{↔}R_{1}{{NH}_{3}}^{+}+R_{1}{NHCOO}^{-}$$
(2)$${CO}_{2}+2R_{1}R_{2}{NH}_{2}\overset{}{↔}R_{1}R_{2}{NH}^{+}+R_{1}R_{2}{NCOO}^{-}$$
(3)$${CO}_{2}+2R_{1}R_{2}R_{3}N+H_{2}O\overset{}{↔}R_{1}R_{2}{NH}^{+}+{{HCO}_{3}}^{-}$$
(where R_1_, R_2_, and R_3_ are aryl/alkyl groups).

However, García-Abuín et al. [32] observed that MEA produced a mixture of carbamate and bicarbonate as the main reaction products during CO_2_ absorption. The reaction starts with the reversible reactions between MEA and CO_2_ to form carbamate at low CO_2_ loading, followed by the CO_2_ hydration to form HCO_3_^−^/CO_3_^2−^ under high CO_2_ loading, and accompanied by the hydrolysis of carbamate. The reaction mechanism of CO_2_ capture into MEA solution with different CO_2_ loadings is shown in Figure 2.

**Table 3 nanomaterials-13-02050-t003:** Comparison between different liquid amines [33,34,35,36,37,38].

Criteria	Alkanolamines			Sterically Hindered Amines
	Primary	Secondary	Tertiary	
**Examples**	Monoethanolamine (MEA)	Diethanolamine (DEA)	N-methyldiethanolamine (MEDA)	2-amino-2-methyl-1- propanol (AMP)
**Structure**	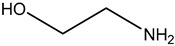	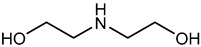	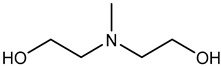	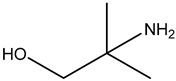
**CO_2_ loading at 59.85 °C** **(mol CO_2_/mol amine)**	0.426 (MEA 30 wt%)	0.404 (DEA 30 wt%)	0.141 (TEA 30 wt%)	0.466 (AMP 30 wt%)
**Regeneration efficiency (%) at 90 °C**	75.5	84.89	95.09	
**Advantages**	Inexpensive solvent Reversible absorption High selectively (between acid and other gases) Reacts with CO_2_ more rapidly	Inexpensive solvent Reversible absorption High selectively (between acid and other gases) Reacts with CO_2_ more rapidly	Inexpensive solvent Reversible absorption High selectively (between acid and other gases) High CO_2_ absorption capacity Requires low regeneration energy	High CO_2_ absorption capacity Requires low regeneration energy
**Disadvantages**	Lower CO_2_ absorption capacity Requires high regeneration energy Oxidative degradation occurs in the presence of other gas components Corrosive High capital costs	Lower CO_2_ absorption capacity Requires high regeneration energy Oxidative degradation occurs in the presence of other gas components Corrosive High capital costs	Reaction rate with CO_2_ is low Compared to MEA and DEA Corrosive High capital costs	Low reaction rate

According to Table 3, there are three categories of alkanolamines that show increased capital costs due to requirement of specialized and expensive materials for construction [28]. On the contrary, degradation of alkanolamine causes operational, and environmental problems including high amount of absorbent required, corrosion of equipment, and demanding of energy [24].

Among three different alkanolamines, MEA is commonly considered as a well-established solvent to separate CO_2_ because it can be regenerated easily [35]. On the other hand, Rinprasertmeechai et al. reported the order of CO_2_ absorption capacity of the different alkanolamines as MEA > DEA > TEA (see Table 3) [33]. Moreover, they have further showed the regeneration ability of the amines in the following order: MEEA > > DEA > MEA. According to Table 3, MEA exhibits high CO_2_ adsorption capacity as it reacts more rapidly with CO_2_ compared to MEDA by forming carbamates. However, MEDA shows high regeneration efficiency and requires lower energy [36]. Moreover, Wang et al. found that, when MEA and MEDA are mixed with the appropriate ratio, the energy consumption for CO_2_ regeneration is reduced significantly [37].

Sterically hindered amines are based on primary or secondary amines with bulky alkyl groups, which is inhibited from reacting with CO_2_ through the effect of steric hindrance [28]. One example of sterically hindered amines is 2-amino-2-methyl-1-propanol (AMP). Steric factor reduces the stability of the formed carbamate due to the weak interaction between the CO_2_ molecule and the NH_2_ group, promoting fast hydrolysis to form bicarbonate and reducing regeneration energy. Due to the immediate regeneration process of AMP, the NH_2_ group can react with CO_2_ molecules over and over, increasing CO_2_ adsorption (see Table 3). Moreover, Dave et al. [38] compared the CO_2_ absorption of different liquid amine classes and showed a lower regeneration energy requirement for 30 wt% AMP over 30% MEA, 30% MEDA, 2.5% NH_3_, and 5% NH_3_ [38].

Recently, ionic liquids (IL) have also been investigated as liquid solvents for CO_2_ capture due to their low vapor pressure, thermal stability, non-toxicity, and adsorption capacity [39,40,41]. The widely studied ILs include bis(trifluoromethylsulfonyl)imide (TF_2_N), tetrafluoroborate (BF_4_), and hexafluorophosphate (PF_6_) [39,40,41]. However, the main drawbacks of the ILs are high viscosity and production high cost.

### 2.4. Comparison between Major Non-Carbonaceous Solid Sorbents for CO_2_ Capture and Importance of Silica Materials

Due to the low contact area between gas and liquid, low CO_2_ loading, and absorbent corrosion associated with liquid amine-based sorbents, solid sorbents for CO_2_ capture have attracted significant attention in recent years [42,43]. Various solid adsorbents have been proposed according to their structures and compositions, adsorption mechanisms, and regeneration process [43]. Many solid sorbents are cheap and readily available and show low heat capacities, fast adsorption kinetics, high CO_2_ adsorption capacities and selectivity, and high thermal, chemical, and mechanical stabilities [43].

Commercially available solid adsorbents for CO_2_ capture include carbonaceous materials such as activated carbons, nanofibrillated cellulose (CFCs), carbon nanotubes (CNTs), and non-carbonaceous materials, including silica, zeolites, hollow fibers, and alumina [6]. These materials show different surface morphologies, pore structures, specific surface areas, and functional groups.

Carbonaceous adsorbents are widely used for CO_2_ capture due to their relative abundance, low cost, renewability, and high thermal stability. However, the weak CO_2_ adsorption capacities of carbonaceous materials at 50–120 °C make it challenging to use in industrial CO_2_ capture [44]. Therefore, much research focus has been given to non-carbonaceous materials. Table 4 tabulates commonly tested non-carbonaceous solid adsorbents for CO_2_ capture and their advantages and setbacks.

As mentioned earlier, carbonaceous adsorbents such as activated carbon have been widely used for CO_2_ capture due to their wide availability, low cost, and high thermal stability. However, weak CO_2_ adsorption of carbonaceous materials in the range of 50–120 °C leads to high sensitivity in temperature and relatively low selectivity in operation [44]. Therefore, many research works have focused on non-carbonaceous materials such as mesoporous silica, and zeolites due to their advantages, as shown in Table 4.

Zeolites are aluminosilicates with ordered three-dimensional (3D) microporous structures with high crystallinity and surface area [44]. The adsorption efficiencies of zeolites are primarily affected by their size, charge density, and chemical composition of cations in their porous structures [37]. It has been reported that the CO_2_ adsorption of zeolites increases as the Si/Al ratio increases and is exchanged with alkali and alkaline-earth cations in the structure of zeolites [45]. However, zeolites present several drawbacks, such as relatively low CO_2_/N_2_ selectivity and high hydrophilicity [46]. Apart from the above, zeolites show reduced CO_2_ adsorption capacity when CO_2_/N_2_ mixtures contain moisture, and zeolites require high temperatures (>300 °C) for regeneration [47].

Recently, metal-organic frameworks (MOFs) have gained much attention owing to their unique properties, such as tunable pore structure and high surface area [48]. However, when exposed to gas mixtures, the MOFs show decreased adsorption capacities [46]. Moreover, previous reports indicate that MOFs are promising materials for CO_2_ capture in laboratory settings; however, further research is required to confirm their practical applicability [49]. Water vapor also negatively affects the application of these sorbents by competing and adsorbing them onto physisorbents, thus decreasing their CO_2_ adsorption capacity [50].

Ordered mesoporous silica materials are good candidates because of their high surface area, high pore volume, tunable pore size, and good thermal and mechanical stability. So far, mesoporous silica includes the families of MCM (Mobil Company Matter: M41S, Santa Barbara Amorphous type material (SBA-n), anionic surfactant-template mesoporous silica (AMS) [44]. However, the CO_2_ adsorption capacities of them observed at atmospheric pressure are not high. Therefore, many studies have been recently reported on the functionalized mesoporous and nanoporous silica for efficient CO_2_ capture [51,52].

Several reviews have recently focused on the potential applications of porous silica materials as CO_2_ adsorbents. Reddy et al. [53] reported CO_2_ adsorption based on porous materials of MOFs, clay-based adsorbents, porous carbon-based materials, and polymer-based adsorbents. Liu et al. [54] also discussed different porous materials, including silica, for post-combustion CO_2_ capture [54]. However, more information on silica-based sorbents and their synthesis methods still needs to be available. Therefore, this review mainly discusses CO_2_ capture onto different porous and functionalized silica materials. In addition, an overview of synthesis processes and a comparison between the adsorption capacities are also profoundly discussed. Finally, the technical challenges and the future research directions of the porous silica materials for CO_2_ adsorption are also presented in this review.

## 3. CO_2_ Capture Methods

Two general mechanisms are involved in CO_2_ capturing using solid sorbents: chemisorption and physisorption. Table 5 represents the major comparison between chemisorption and physisorption. However, the two mechanisms differ in the interactions between the gas molecules and the sorbent surface. During chemisorption, gas molecules are chemically bonded to the surface, whereas in physisorption, there is no chemical binding of the gas molecules to the surface, see Figure 3.

CO_2_ capturing using solid adsorbent is a selective separation [24]. The critical parameters for solid sorbents are surface tension, pore size, temperature, and pressure [24,59]. The adsorption process involves repeated cycles of adsorption and desorption, also known as regeneration. The four main adsorption processes are: (i) Pressure Swing Adsorption (PSA), (ii) Temperature Swing Adsorption (TSA), (iii) Electric Swing Adsorption (ESA), and (iv) Vacuum Swing Adsorption (VSA). Figure 4 shows the four different adsorption processes and their unique characteristics.

In the PSA process, adsorption happens at low pressure, and desorption occurs at high pressure. The adsorption of the TSA process occurs in the temperature range of 40–120 °C and the desorption process in the temperature range of 120–360 °C, respectively [3]. The VSA process involves CO_2_ uptake at ambient pressure, then swings to a vacuum condition to regenerate the adsorbent. The ESA process conducts the adsorption–desorption process by changing the electrical supply [3]. Activated carbons, MOF, zeolites, activated alumina, and silica gel are mainly used sorbents in TSA and PSA processes, while ESA is considered less costly compared to those of both TSA and VSA [59].

The microwave-swing adsorption (MWSA) is another adsorption process that has recently received considerable attention due to its efficient energy management. Unlike in conventional heating, where solids heat through conduction and convection, the MWSA process can transfer energy directly to the adsorbate without transferring the heat to both the adsorbate and adsorbent [11,60].

## 4. CO_2_ Adsorption Using Mesoporous Silica Materials (Physisorbents)

### 4.1. Mesoporous Silica Materials

Mesoporous silica materials are used for various applications, including catalysis and wastewater treatment [61]. Mesoporous silica has unique properties such as uniformity of pore distribution (with size between 0.7 and 50 nm), high surface area (around 1000 m^2^/g), and good thermal stability [62]. The first synthesized mesoporous silica material was M41S in the 1990s [63]. However, the development of surfactants and synthesis protocols have been able to prepare many types of mesoporous silicas such as MCM-41, SBA-15, SBA-16, FDU-2, MCM-50, and KIT-5 with a diverse range of pore geometries such as cubic, and hexagonal, and morphologies such as rods, spheres, and discs [64].

In 1990, Mobil Oil Corporation discovered molecular sieves of the M41S family consisting of silicate/aluminosilicate [65]. Typically, these materials are prepared via the sol-gel method. Three well-defined structural arrangements have been identified after studying the effect of surfactant concentration, and those are hexagonal (MCM-41), cubic (MCM-48), and lamellar (MCM-50) structures. Therefore, these materials (M41S family) exhibit mesoporous arrays with amorphous walls of about 10 Å (1 nm) [65]. Moreover, the structural ordering of these M41S family materials can be changed with increasing hydrothermal synthesis temperature and time [65]. These M41S molecular sieves are mainly applied in catalysis [66], adsorption [65], and controlled release of drugs [67]. The main advantage of this mesoporous silica is its unique chemical structure consisting of the high density of functional silanol groups (Si–OH), pore size and shape can be molded during the synthesis process, and the internal surface can be easily modified with organic and inorganic groups [65,68,69].

Santa Barbara Amorphous family (SBA) first prepared silica-based materials with well-ordered mesoporous in 1998 [65]. This material group consists of SBA-2 (hexagonal close-packed array), SBA-12 (three-dimensional hexagonal network), SBA-14 (cubic structure), SBA-15 (two-dimensional hexagonal), and SBA-16 (structured in a cubic cage) [65,70]. These nanostructured mesoporous materials comprise a silica-based framework with uniform and well-ordered mesopores, large pores, thick and porous walls, high surface area, and high thermal stability [69,71]. The most widely investigated members of the SBA-n family in the literature are SBA-15 and SBA-16. The SBA-15- and SBA-16-based mesoporous arrays are commonly utilized as adsorbents [69], catalysts or catalytic [72], and drug deliveries [73].

The Fudan University synthesized mesoporous materials family (FDU-n)-based mesoporous silica arrays with well-ordered mesostructures and pore arrangements, high surface area, large and uniform distribution of pore diameter, amorphous pore-wall structures, and thermal and mechanical stability [74]. FDU-1-based mesoporous materials have a 3D face-centered cubic (FCC) structure with large cage-like mesopores, while FDU-2 mesoporous array possesses a mesostructured FCC unit cell and well-ordered 3D architecture [69].

On the contrary, the mesoporous material series of the KIT-n family, where n = 1, 5, or 6, are mainly represented by the KIT-1, KIT-5, and KIT-6. However, KIT-1-based mesoporous silicas exhibit a 3D architecture in a disordered framework with high surface area, large pore volume and pore diameter, and thermal and hydrothermal stability [75]. KIT-5-based nanostructured mesoporous materials have well-ordered 3D cage-like mesopores in a face-centered close-packed cubic lattice architecture [69]. In addition, KIT-6 shows 3D mesoporous amorphous walls with large pore size, uniform pore distribution, high surface area, and thermal stability [69].

Moreover, mesoporous silica materials of the M41S, SBA-n, FDU-n, and KIT-n families are used in a wide range of applications such as separation, catalysis, drug release adsorption, sensors, matrix solid-phase dispersion (MSPD) and solid-phase extraction [69].

### 4.2. Synthesis Procedures of Mesoporous Silica

Initially, Stöber et al. [76] discovered an effective method for synthesizing monodispersed silica particles. This process consists of hydrolysis of tetraethyl orthosilicate (TEOS) using ammonia as a catalyst in water and ethanol solution. This method leads to the synthesis of silica particles [77]. In this reaction, TEOS undergoes hydrolysis in an ethanol/ammonia solution. As a result, it produces silanol monomer (-Si-OH) with the epoxy groups (-Si-OEt), as shown in Equation (4). Then silanol groups undergo condensation to produce branched siloxane clusters, which causes to initiate the nucleation and growth of silica particles, see Equation (5). Simultaneously, silanol monomers react with the unhydrolyzed TEOS via condensation (see Equation (6)) and participate in the nucleation and growth of silica particles [30]. Moreover, the particle size of Stöber silica depends on the concentration of the aqueous ammonia solution and water in the ethanol reaction [30].
(4)$${Si(OEt)}_{4}+XH_{2}O\overset{Hydrolysis}{\to }{SiO(OEt)}_{4-x}{\left(OH\right)}_{x}+XEtOH$$
(5)$${SiO\left(OEt\right)}_{4-x}{\left(OH\right)}_{x}\overset{Condensation}{\to }{\left(OEt\right)}_{4-2x}{\left(OH\right)}_{2x-2}+H_{2}O$$
(6)$${Si\left(OEt\right)}_{4}+{SiO\left(OEt\right)}_{4-x}{\left(OH\right)}_{x}\overset{Condensation}{\to }{\left(OEt\right)}_{7-x}{\left(OH\right)}_{x-1}+EtOH$$

Many experimental factors control hydrolysis, silica condensation rate, assembly kinetics, nucleation, and growth rates [65,78]. The pH is an essential factor that influences the charges of silica species. Rates of hydrolysis of silane and condensation of the siloxane bond depend strongly on the charge states. Hydrolysis of the Si–OR bond in silanes could be catalyzed by acid and base conditions, but its rate is prolonged near the neutral conditions [78].

Sakamoto et al. [79] prepared silica nanoparticles (NPs) via the evaporation and self-assembly of silicate and quaternarytrialkylmethylammonium as a surfactant. This study shows that the size of NPs depends on the ratio between the surfactant and silica precursor. Apart from that, Sihler et al. [80] used dye-stabilized emulsion to synthesize SiO_2_ NPs. Moreover, this synthesis method provides silica capsules and sub-particles with precise size control. Monodispersed colloidal silica NPs (diameter of 15–25 nm) were prepared by Murray et al. [81]. In this study, as the silica source, octadecyltrimethoxysilane (OTMS) was used.

Simple synthesis methods called soft and hard templating are also applied to increase the pore volume and loading capacity of prepared hollow mesoporous SiO_2_ [82]. Template synthesis of mesoporous materials typically enrolls in three steps: template preparation, template-directed synthesis of the target materials using sol-gel, precipitation, hydrothermal synthesis, and template removal [83,84].

The hard-templating method involves nano-casting using pre-synthesized mesoporous solids [85]. Hard templating is a facile synthesis method for fabricating porous materials with a stable porous structure. The structure replication is very straightforward [83]. This approach utilizes porous hard templates such as mesoporous silica. The pores of these templates are impregnated with a precursor compound for the desired product, which is then thermally converted into the product. The template is finally removed to yield the desired mesoporous material as a negative structural replica of the hard template [83]. However, the method is costly and time-consuming. Moreover, the mesoporous parameters, such as mesostructure and pore sizes, are difficult to change [84].

In contrast, soft templating methods use cationic and anionic surfactants or block copolymers as templates [78]. During the synthesis, surfactant or block copolymers are used as a soft template. Moreover, the increase in surfactant micelle concentration causes the formation of a large assembly or self-assembly of 3D mesoporous [30]. Different 3D micelle structures can be obtained by varying the solvent ratio between the aqueous and non-aqueous and adding co-solvents. Moreover, the silica source interacts with the structure-directing agent (SDA) without any phase separation. The interactions between ions or charged molecules are vital in forming well-defined porous nanostructures [85].

The soft templating method mainly depends on the self-assembly of the surfactant [83]. The process is based on the interactions between inorganics. The mesoporous structure of the final material is obtained after the removal of the pore-templating surfactant or block copolymers by low-temperature calcination (up to 600 °C) or by different washing techniques (extraction) [83]. Figure 5 represents the synthesis mechanism of mesoporous silica in the presence of a cationic surfactant. The synthesis process of mesoporous silica is carried out using TEOS as the silica source [30]. In this process, surfactant plays a significant role in defining the pore size and volume of silica [30]. Cationic surfactant forms micelle structures with water, which arranges the cationic “heads” of the surfactant molecules to the outer side. It resulted in the hydrophobic “tails” collected in the center of each micelle. As the next step, silica molecules cover the micelle surface. Finally, the surfactant is removed via calcination or extraction, and it results in porous silica [30,86,87].

Figure 6 shows the schematic diagram for synthesizing mesoporous silica using block copolymer. As can be seen from Figure 6, titania-incorporated organosilica-mesostructures (Ti-MO) are synthesized via condensation method using silica precursors ([3-(trimethoxysilyl) propyl] isocyanurate and tetraethylorthosilicate) and titanium precursor (titanium isopropoxide) in the presence of the triblock copolymer, Pluronic P123 [88]. This method consists of template removal using two independent steps (i) extraction with a 95% ethanol solution and (ii) calcination of the sample at 350 °C. This method improves the adsorption capacity and enhances the structural properties such as specific surface area, micro-porosity, and pore volume.

**Figure 5 nanomaterials-13-02050-f005:**
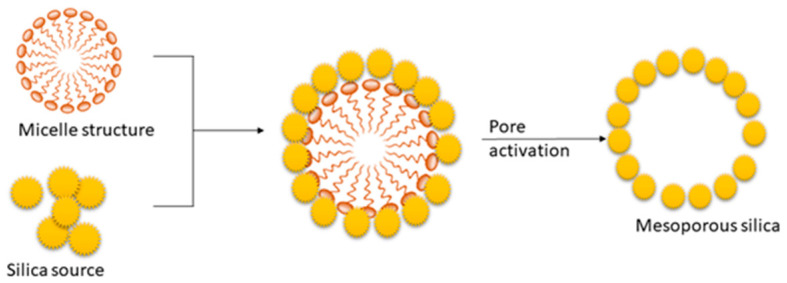
Mechanism for the synthesis of mesoporous silica in the presence of a cationic surfactant (Reprinted with permission from Kim et al. [89]).

The synthesis of MCM-41 and SBA-15 is performed using cetrimoniumbromide (CTAB) and Pluronic P123 surfactant. The CTAB is an ionic surfactant and acts as stearidonic acid (SDA) and which causes the formation of a hexagonal array of mesostructured composites [12]. However, as the final step, surfactants are removed by heating in air at high temperatures or by solvent extraction to obtain MCM-41 and SBA-15 [30]. Wu et al. [79] and Hao et al. [88] reported a detailed description of the mechanism. Paneka and co-workers have reported the synthesis of MCM-41 from fly ash using a hydrothermal process. However, the synthesis of MCM-41 shows reduced BET surface area, increased pore volume, and pore size [89].

Recently, Singh and Polshettiwar [90] reported the synthesis of silica nano-sheets using ammonium hydroxide. They have developed a method to synthesize silica nano-sheets using lamellar micelles as soft templates in a water-cyclohexane solvent mixture. Zhang et al. [19] also reported the large-scale synthesis of mesoporous silica nanoparticles. Reported data show that various morphologies and particle sizes have been obtained during the synthesis. For synthesis process, the reaction occurred at atmospheric pressure with a sol–gel technique using CTAB as a template.

### 4.3. Importance of Micro-Porosity and CO_2_ Adsorption Capacity of Mesoporous Silica Materials

The textural properties, including surface area, pore diameter and volume of mesoporous materials, are usually measured by studying nitrogen adsorption–desorption isotherms. The specific surface area is calculated using the volume adsorbed at different relative pressure data by the Brunauer–Emmett–Teller (BET) method [65]. Apart from that, the pore volume and pore size distribution are determined using the Barrett–Joyner–Halenda (BJH) method [65].

Furthermore, the textural properties are important parameters when considering CO_2_ adsorption using physisorbents. Moreover, microporosity plays a major role in CO_2_ gas adsorption because it involves the diffusion of CO_2_ molecules into the physisorbent [91,92,93]. Table 6 represents the textural properties and CO_2_ absorption capacity recorded for different ordered mesoporous silica materials studied.

MCM-41 has high porosity and an ordered hexagonal pore structure arrangement. However, it showed a low CO_2_ adsorption capacity of 0.63 mmol/g at 25 °C and 1 bar (see Table 6). This behavior may be due to the weak interactions between the hydroxyl groups of MCM-41 and CO_2_ molecules [93]. Son et al. prepared KIT-6, SBA-15, SBA-16, MCM-48, and MCM-41 and their textural properties of the materials are tabulated in Table 6 [94]. The pore size of mesoporous materials varied in the descending order of KIT-6 > SBA-15 > SBA-16 > MCM-48 > MCM-41. The KIT-6 exhibited the largest pore volume among the other sorbents. These combined features of large pore size and large pore volume would enable KIT-6 to better accommodate the bulky polyethyleneimine (PEI) with little hindrance, allowing higher loadings inside silica particles than other silica-supported materials. Moreover, Zelěnák and co-workers prepared three mesoporous silica materials with different pore sizes (3.3 nm MCM-41; 3.8 nm SBA-12; 7.1 nm SBA-15) [95]. During their studies, amine functionalization was investigated with the effect of pore size and architecture on CO_2_ sorption. According to the data, SBA-15 showed the highest CO_2_ adsorption of 1.5 mmol/g due to the highest amine surface density in SBA-15 [95].

Lashaki and Sayari [96] also investigated the impact of the support pore structure on the CO_2_ adsorption performance of SBA-15 silica. In this study, SBA-15 silica supports were used to obtain different pore sizes and intra-wall pore volumes. These materials were functionalized further with triamine through dry and wet grafting. CO_2_ sorption measurements showed the positive impact of support with large pore size and high intra-wall pore volume on adsorptive properties, with the former being dominant. Large pore volume influenced the load of more amine groups, CO_2_ uptakes, and CO_2_/N_2_ ratios and faster kinetics. When the intra-wall pore volume decreased by 53%, it caused a reduction in CO_2_ uptake capacity by up to 63% and CO_2_/N_2_ ratios by up to 62% and slower adsorption kinetics. Moreover, it was inferred that large pore size and high intra-wall pore volume of the support improved the adsorptive properties via enhanced amine accessibility [96].

**Table 6 nanomaterials-13-02050-t006:** The textural properties and CO_2_ absorption capacity of various ordered mesoporous silica materials.

Types of Mesoporous Silica	Mesostructure	Silica Source	Surfactant/ Block Co-Polymer	BET Specific Surface Area (m^2^/g)	Pore Volume (cm^3^/g)	Pore Size (nm)	Adsorption Capacity (mmol/g)	Adsorption Conditions		Ref.
								Temp. (°C)	Pressure (Bar)	
**KIT-5**	3D-cubic	TEOS	Pluronic P123	711	1.05	8.04	0.48	30	1	[97]
**KIT-6**	3D-cubic	TEOS	Pluronic P123	895	1.22	6.0	-	-	-	[94]
**MCM-41**	Hexagonal	Na_2_SiO_3_	CTAB	994	1.00	3.03	0.63	25	1	[93]
		Na_2_SiO_3_	CTAB	993	1.00	3.1	0.63	25	1	[98]
		Na_2_SiO_3_	CTAB	980	0.92	4.08				[90]
**MCM 48**	Cubic	SiO_2_	CTAB	1287	1.1	3.5		25	1	[99]
**SBA-15**	2D hexagonal	TEOS	P123	1254	2.44	11.4	-	-	-	[100]
**SBA-16**	Cubic cage	TEOS	Pluronic F127	736	0.75	4.1	-	-	-	[94]
**SNS**		TEOS	Pluronic F127	394	0.10	21.1	2.06	25	1	[101]
**SNT**		TEOS	Pluronic F127	319	0.07	26.0	2.46	25	1	[101]

Where CTAB: cetyltrimethylammoniumbromide and hexadecyltrimethylammoniumbromide, F127: tri-block copolymer F127, Na_2_SiO_3_: sodium silicate, P123: triblock copolymer (Pluronic P123), SiO_2_: silica, SNS: silica nano spheres, SNT: silica nano tube, TEOS: tetraethyl orthosilicate.

## 5. Chemisorbents (Amine Functionalized Si-Based Materials)—Application at Low and High Temperature CO_2_ Sorption

In physisorption, CO_2_ molecules attach to the pore walls through weak Van der Waals and pole–pole interactions [102]. However, the unmatched pore size of the mesoporous silica and the small diameter of the CO_2_ gas molecule causes low CO_2_ adsorption capacities. The heat of adsorption of the physisorption process ranges from −25 to −40 kJ/mol [103], which is approximately closer to the heat of sublimation [104]. Recently, it has been reported about mesoporous silica materials with improved CO_2_ sorption capacity with amine functionalization [105]. Hence, the adsorption capacity of CO_2_ depends on the nature of the amine groups and the spacing between the amino silanes [106]. Figure 7 represents the different types of amino silanes and polymer-containing amino groups used during the functionalization of mesoporous silica for enhanced adsorption or separation.

### 5.1. Synthesis of Amine-Functionalized Silica

Amine-based adsorbents are generally synthesized using three approaches: the selection of solid scaffolds with high amine loading ability, use of amines with high nitrogen content, and use of effective methods for introducing amine groups [44]. Synthesis methods of amine-functionalized silica materials include three main pathways: impregnation, grafting, and in-situ polymerization. Figure 8 shows the three different synthesis processes of amine-functionalized silica materials.

In impregnation, amines are physically trapped in the pores of silica materials. Moreover, the performance of amine-silica adsorbents is influenced by the pore structure of silica. For example, Chen et al. [107,108] reported that the CO_2_ adsorption capacity decreases as the pore diameter decreases. Moreover, surfactants, surface functional groups, amine types and heteroatom incorporation affect the impregnation process [54]. In this method, the amine loading is also influenced by the total pore volume of the silica materials and the amine density.

Moreover, if the amount of amine exceeds the capacity of the support, the amine species agglomerate on the support. The main advantage of this method is the simplicity and easy synthesis procedure. Further, many amine species can be incorporated with mesoporous silica due to the large pore volume of the porous silica materials [109].

Grafting occurs between an aminosilane and silica, as shown in Figure 8, where amine groups are grafted on the silica surface via covalent bonds [110]. Mainly, three methods are used for grafting amine onto silica support: post-synthesis grafting, direct synthesis by co-condensation (one-pot synthesis), and anionic template synthesis [111]. In a typical process, silica is dispersed in a solvent, amino silanes are added, and the mixture is heated under reflux. However, the amount of amine incorporated is related to the number of hydroxyl groups on the silica surface [109]. In-situ polymerization is another promising method for functionalizing porous silica, such as hyperbranched aminosilica (HAS). This category of supported sorbents can be considered a hybrid of grafting and impregnation [112].

**Figure 8 nanomaterials-13-02050-f008:**
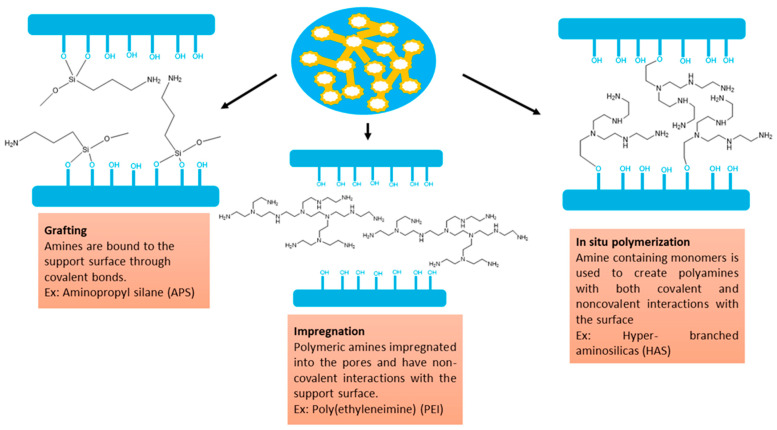
Different types of synthesis processes of amine-functionalized silica materials (Schematic shows supported amines (yellow) in the pores (blue)) (Reprinted with permission from Bollini et al. [113]).

Solvents, including toluene are also used for grafting. Moreover, the impregnating technique is widely employed because of its simplicity, low cost, environmental friendliness, and convenience for large-scale production [114]. However, to overcome the challenges caused by grafting, researchers have recently investigated aminosilane gas-phase grafting and supercritical fluid impregnation [115].

Supercritical fluid impregnation is one of the most effective, simple, and reproducible methods for producing homogeneous, covalently bonded, and high-density silane [115]. López-Aranguren et al. [115] synthesized functionalized silica via supercritical CO_2_ grafting of aminosilanes. This study used silica gels (4.1 and 8.8 nm pore diameter), mesoporous silica MCM-41 (3.8 nm pore diameter), and mono- and di-aminotrialkoxysilane.

The double-functionalization method of mesoporous materials is also widely used in recent years. Several studies prepared amine–silica composites using the double-functionalization method [116,117,118]. Those studies employed impregnation and grafting to improve CO_2_ uptake [116].

### 5.2. Comparison of Adsorption Capacities of Silica-Based Sorbents

Nigar et al. [99] synthesized the ordered mesoporous (MCM-48) silica with different silane molecules, including 3triethoxysilylpropylamine, 3-(2-aminoethylamino) propyl] trimethoxysilane and 2-[2-(3-trimethoxysilylpropylamino)ethylamino]ethylamine. Here-in, silane groups were covalently bound with the silica groups, as shown in Figure 9. The functionalization caused the reduction in the surface area and the pore volume compared to the non-functionalized MCM-48 (1287 m^2^/g and 1.1 cm^2^/g) (see Table 7). Most importantly, it is seen that the increment of the number of amine groups in silane molecules leads to a decrease in CO_2_ absorption capacity governed via chemisorption [99].

Moreover, Park et al. [29] synthesized functionalized silica using silane molecules, similar to the study conducted by Niger et al. [99]. However, they compared in-situ polymerization and grafting. According to the data (see Table 7), the sorbent prepared through in-situ polymerization shows enhanced CO_2_ adsorption capacity. Ahmed et al. [93] reported a detailed study about the functionalization of mesoporous Si-MCM-41 with different loadings of PEI. According to their work, with increasing PEI loading, the CO_2_ adsorption capacity also increased (see Table 7). They mentioned that the enhanced adsorption is due to branched PEI with many amino groups, providing potential sites for CO_2_ molecules. Moreover, the hierarchical mesoporous structure of Si-MCM-41 made these sites accessible to CO_2_ by improving the dispersion of PEI [119].

Gargiulo and co-workers investigated the effect of temperature on CO_2_ adsorption capacity on SBA-15 and PEI. CO_2_ adsorption was evaluated at 25, 40, 55, and 75 °C temperatures [120]. The experimental data showed a significant dependence of the CO_2_ adsorption capacity on temperature (Table 7). The effect of pore dimension on CO_2_ adsorption over amine-modified mesoporous silicas was reported by Heydari-Gorji et al. [100]. The pore lengths of the silica supports were 25, 1.5, and 0.2 μm. It showed that the small pore size of silica materials exhibited the highest adsorption capacities due to the enhanced amine accessibility inside the pores. Heydari-Gorji and Sayari [121] showed PEI impregnation for CO_2_ removal applications. They demonstrated that PEI-functionalized silica materials were thermally stable at mild temperatures. Kuwahara et al. [122] synthesized poly(ethyleneimine)/silica composite adsorbents by incorporating zirconium (Zr) into the silica support. The authors observed Zr sites with increased CO_2_ adsorbent capacity (see Table 7), regeneration, and stability.

Apart from that, Kishor and Ghoshal [123] investigated the effects of the structural parameters such as pore size, pore volume, and surface area of the silicas and amine-functionalized silica on the CO_2_ sorption capacity. The authors used various silica materials such as KIT-6, MCM-41, SBA-15, and HV-MCM-41. The wet impregnation method was employed to prepare the pentaethylenehexamine (PEHA) functionalized silica. The CO_2_ capture capacities of the amine-functionalized silicas were measured at 105 °C and 1 bar pressure conditions (see Table 7). The KIT-6 showed the highest CO_2_ capture capacity of 4.48 mmol/g of CO_2_ at 105 °C and 1 bar pressure) among all the sorbents investigated (MCM-41 < HVMCM-41 < SBA-15 < KIT-6). Furthermore, KIT-6 showed enhanced amine density distribution due to large pore volume. All the other silica sorbents remained stable up to ten adsorption–desorption cycles.

**Table 7 nanomaterials-13-02050-t007:** CO_2_ adsorption capacities and structural properties of amine functionalized silica-based adsorbents.

Silica-Based Sorbent	Amine Types	CO_2_ Adsorption Performance Capacity (mmol/g)	Conditions		BET Specific Surface Area (m^2^/g)	Pore Volume (cm^3^/g)	Pore Size (nm)	Preparation Methods	Ref.
			Temperature (°C)	Pressure (Bar)					
**DWSNT**	-	0.1	25		83	0.58		Immobilization	[124]
**DWSNT**	APTMS	1.0	25		112	0.72		Immobilization	[124]
**DWSNT**	MAPTMS	1.5	25		114	0.79		Immobilization	[124]
**DWSNT**	DEAPTMS	1.8	25		68.9	0.49		Immobilization	[124]
**DWSNT**	AEAPTMS	2.25	25		60.9	0.45		Immobilization	[124]
**HAS**	Aziridines	3.25	25		71	5	0.15		[125]
**HPS**	PEI	2.44	75	1	0.5	0.009		Impregnation	[126]
**HVMCM-41**	PEHA	4.07	105	1				Impregnation	[123]
**KIT-6**	PEHA	4.48	105	1				Impregnation	[123]
**MCM-41**	EDA	1.19	35					Impregnation	[127]
**MCM-41**	DETA	1.43	35					Impregnation	[127]
**MCM-41**	TEPA	1.96	35					Impregnation	[127]
**MCM-41**	PEHA	2.34	35					Impregnation	[127]
**MCM-41**	MEA (3%)	11.39	25		426	0.42	3.12	Impregnation	[128]
**MCM-41**	PEI	0.39	40	0.15	443	0.340	2.95	Impregnation	[49]
**MCM-41**	PEI	0.22	75	1	590	1.4	13.6	Impregnation	[120]
**MCM-41**	PEI Aziridine	0.98	75	1				In-situ grafted polymerization	[129]
**MCM-41**	APTS	94	25	1	10	0.01		Grafting	[114]
**MCM-41**	APTS	0.70	30	0.1					[130]
**MCM-41**	APTS	2.48	20	1	17	0.04	20.1	Grafting	[131]
**MCM-41**	PEHA	4.5	105	1				Impregnation	[120]
**MCM-41**	MEA	0.89	25	1	19	0.82		Impregnation	[98]
**MCM-41**	DEA	0.80	25	1	13	0.07		Impregnation	[98]
**MCM-41**	TEA	0.63	25	1	213	0.17		Impregnation	[98]
**MCM-41**	Branched PEI	1.08	100	1	6	0	-	Impregnation	[93]
**MCM-41**	Branched PEI	0.79	100	1	12	0.04	-	Impregnation	[93]
**MCM-41**	Branched PEI—(30 wt%)	0.70	100	1	80	0.14	-	Impregnation	[93]
**MCM-41**	Branched PEI	28	100	1	104	0.12	2.05	Impregnation	[93]
**MCM-41**	Branched PEI	17.5	100	1	291	0.17	2.05	Impregnation	[93]
**MCM-41**	TEPA	1.24	25	1	11	0.05	1.8	Impregnation	[132]
**MCM-48**	APTES	0.62	25	1.01	1072	0.52	2.9	Grafting	[99]
**MCM-48**	TRI	0.46	25	1.01	698	0.39	2.6	Grafting	[99]
**MCM-48**	TRI	0.44	25	1.01	463	0.23	2.5	Grafting	[99]
**MSiNTs**	PEI	2.75	92		52.4	0.17	12.4	Impregnation	[133]
**OMS**	PEI	1.4	25		352	0.79		Grafting	[120]
**SAB-15**	PEHA	4.0	105	1				Impregnation	[123]
**SBA-15**	PEI	0.65	25		683	1.19	8.5	Impregnation	[122]
**SBA-15**	PEI/Zr4	1.34	25		642	1.08	8.6	Impregnation	[122]
**SBA-15**	PEI/Zr7	1.56	25		674	1.23	9.5	Impregnation	[122]
**SBA-15**	PEI/Zr14	1.41	25		601	0.69	7.0	Impregnation	[122]
**SBA-15**	PEI/Ti1.4	0.24	25		510	0.39	4.4	Impregnation	[122]
**SBA-15**	NH_2_OH	1.65	25	1	435.6	0.54	6.85	Grafting	[134]
**SBA-15**	APTMS	1.46	25	0.15	82	0.16	5	Grafting	[135]
**SBA-15**	TEPA	2.45	70		5	0.03		Grafting	[100]
**SBA-15**	AMP	1.79	70		372	0.21		Grafting	[120]
**SBA-15** **(0.2 µm)**	PEI	5.84	100	1	590	1.44	13.6	Impregnation	[120]
**SBA-15 (1.5 µm)**	PEI	-	100	1	746	0.80	7.2	Impregnation	[120]
**SBA-15 (25 µm)**	PEI	5.81	100	1	580	0.95	10.5	Impregnation	[120]
**SiO_2_**	APTES	4.3	30		67	0.51		In-situ polymerization	[29]
**SiO_2_**	AEAPTMS	5.7	30		45	0.37		In-situ polymerization	[29]
**SiO_2_**	TRI	5.6	30		25	0.22		In-situ polymerization	[29]
**SiO_2_**	APTES	0.5	30		216	1.11		Grafting	[29]
**SiO_2_**	AEAPTMS	0.3	30		206	1.10		Grafting	[29]
**SiO_2_**	TRI	0.8	30		172	0.99		Grafting	[29]
**SMCM-41**	MEA	10.40	25		405	0.39	3.01	Impregnation	[128]
**SBA-15**	TEPA	4.5	75	1	121.1	0.327		Impregnation	[136]
**MPSM**	TEA	4.27	75	1	34	0.08	9.5	Impregnation	[50]
**MCM-41**	TRI	1.74	25	0.05	678.3	1.47		Grafting	[137]
**MCM-41**	APTES	1.20	30	1	1045.21	2.59	30	Grafting	[138]
**MCM-41**	PEI	0.98	30	1	6.6	0.01	0.8	Grafting	[139]
**MCM-41**	PEI	4.68	45	1	894	1.28	5.1	Grafting	[116]
**MCM-41**	PEI	2.92	50	0.1	508	0.98	2.54	Impregnation	[140]
**MCM-41**	TEPA	2.25	50	0.1	431	0.83	2.21	Impregnation	[140]
**MCM-41-KOH**	PEI-	3.38	50	0.1	391	1.08	2.33	Impregnation	[140]
**MCM-41-Ca(OH)_2_**	PEI-	3.81	50	0.1	411	1.12	2.50	Impregnation	[140]
**MCM-41-CsOH**	PEI-	5.02	50	0.1	306	0.91	2.14	Impregnation	[140]
**MCM-41-KOH**	TEPA-	3.93	50	0.1	322	0.97	2.15	Impregnation	[140]
**MCM-41-Ca(OH)_2_**	TEPA-	3.76	50	0.1	405	0.94	2.31	Impregnation	[140]
**PET-CsOH**	TEPA-	5.42	50	0.1	293	0.97	2.61	Impregnation	[140]
**MCM 48**	PEI	1.09	80	0.24	79.3	0.02	1.68	Impregnation	[141]
**MCM-41**	PEI	1.23	80	0.24	59.1	0.02	1.80	Impregnation	[141]
**SBA-15**	PEI	1.07	80	0.24	62.1	0.01	5.2	Impregnation	[141]
**SBA-15**	PEI	1.77	0	1	783	0.03	7.0	Impregnation	[142]
**SBA-15**	PEI	1.26	45	0.15	399	0.79	8.2	Impregnation	[143]
**MCM 41**	PEI	3.53	25	1	24	0.012		Impregnation	[144]
**MCM 41**	APTS	2.41	25	1	736	0.37		Grafting	[144]
**SBA-15**	PEI	1.84	25	1.2	195	0.39	7.0	Grafting	[145]
**SBA-15-APES**		1.78	25	1.2	190	0.37	7.2	Grafting	[145]
**SBA-15-APES**	PEI	1.54	25	1.2	24	0.21	2.7	Grafting	[145]
**OMS**	PEI	2.43	25	1.2	167	0.33	7.6	Grafting	[145]
**OMS-APES**		3.03	25	1.2	180	0.37	7.2	Grafting	[145]
**OMS-APES**	PEI	1.18	25	1.2	39	0.18	2.3	Grafting	[145]
**OMS-NCC**	Amidoxime	5.54	120	1	315	0.69	9.3		[146]
**MPS-MCC ***		2.41	120		302	0.44	7.0		[147]
**MPS-MCC ****		3.85	120		285	0.40	6.7		[147]
**OMS-MgO**		4.71	120	1	261	0.48	7.25		[148]
**OMS-CaO**		3.85	120	1	163	0.25	6.76		[148]
**SiO_2_-Al_2_O_3_**	APTS	2.64	25	1	740	1.24	5.1	Grafting	[149]
**SiO_2_-Al(NO_3_)_3_**	APTS	0.78	25	1	319	0.63	2.9	Grafting	[149]
**OMS-Ti**		0.81	25	1	487				[88]
**MSiNTs**	APTES	2.87	25	1.2	293	0.79	22	Grafting	[101]
**SNS**	APTES	2.13	25	1.2	210	0.31	19.6	Grafting	[101]
**Al(NO_3_)_3_**	AP	0.98	25	1	359	0.62	10.0		[150]
**OMS-Al-Zr**		2.60	60	1	441	0.61	6.9		[151]

Where, ** MCC-mesoporous silica with amidoxime functionalities, * MCC-mesoporous silica with cyanopropyl groups, APTMS: 3-[2-(2-aminoethylamino)ethylamino]propyltrimethoxysilane, AEAPTMS: [3-(2-aminoethyl) aminopropyl]trimethoxysilane, AMP: 2-amino-2-methyl-1-propanol, AP: 3-aminopropyltriethoxysilane, APTMS: (3-aminopropyl) trimethoxysilane, APTS: 3-aminopropyltrimethoxysilane, DEA: diethanolamine, DEAPTMS: [3-(diethylamino) propyl]trimethoxysilane, DETA: diethylenetriamine, DWSNT: double-walled silica nano tube, EDA: ethylenediamine, HPS: Hierarchically porous silica, MAPTMS: [3-(methylamino) propyl]trimethoxysilane, MCC: microcrystalline cellulose, MEA: monoethanolamine, MPSM: monodispersed porous silica microspheres, MSiNTs: mesoporous silica nanotubes, NCC: nanocrystalline cellulose, OMS: ordered mesoporous organosilica, OMS: Oxide-templated silica monoliths, PEHA: pentaethylenehexamine, PEI: polyethylenimine, SNS: silica nano spheres, TEA: triethanolamine, TEPA: tetraethylenepentamine, TRI: 3-[2-(2-Aminoethylamino)ethylamino]propyltrimethoxysilane.

Sim and co-workers [145] studied the CO_2_ absorption capacity of the silica-based composites papered using SBA-15 and organosilica as silica precursors and N-[3-(trimethoxysilyl)propyl]ethylenediamine as an aminosilane precursor. Herein, PEI was grafted to the silica composites. Results exhibited that organosilica composites (see Table 7) showed the highest CO_2_ adsorption capacity, selectivity, and reproducibility. Another silica composite was prepared by Dassanayake et al. [146] using nanocrystalline cellulose (NCC) and reported that their NCC/mesoporous silica composite showed high CO_2_ absorption capacity (see Table 7), recyclability and thermal stability. Gunathilake et al. [147] synthesized microcrystalline cellulose (MCC) mesoporous silica composites using two MCC-mesoporous silica composites: MCC-mesoporous silica with cyanopropyl groups and MCC mesoporous silica amidoxime groups. CO_2_ adsorption was evaluated at 25 and 120 °C. According to the results, MCC-mesoporous silica with amidoxime functionalities exhibited the highest absorption capacity (see Table 7) at 120 °C due to the oxime and amine groups in amidoxime and hydroxyl groups in MCC which serve as active sites. 

Rao et al. [144] determined the effect of impregnation and grafting of the amine-functionalized MCM-41. The results showed (see Table 7) grafted sorbents with higher thermal stability than the impregnation ones. They concluded that adsorbents modified by impregnation exhibited higher amine-loading efficiencies and, thus, higher CO_2_ adsorption capacities, whereas those prepared by grafting had better thermal and cyclic stability.

Moreover, Tang and co-workers have investigated the effect of inorganic alkalis such as (KOH, Ca(OH)_2_ and CsOH) on the CO_2_ absorption capacity [140]. The results showed that all three kinds of inorganic alkali-containing adsorbents exhibited higher CO_2_ adsorption capacities than tetraethylenepentamine (TEPA) and PEI-modified samples (see Table 7). This may be due to the introduction of inorganic alkali, which changes the chemical adsorption mechanism between adsorbate-CO_2_ and the adsorbent surface due to more hydroxyl groups. Moreover, they reported that CO_2_ adsorption capacities have a linear dependency with the amounts of alkali adsorbents. Apart from that, Gunathilake and Jaroniec [148] reported the incorporation of magnesium oxide (MgO) and calcium oxide (CaO) into mesoporous silica surface (OMS) and applied those materials for CO_2_ sorption at ambient and elevated temperatures. The materials were synthesized using the sol–gel method. However, composite sorbents performed relatively high adsorption capacities (see Table 7). It suggested that MgO and CaO enhanced CO_2_ adsorption via physisorption and chemisorption. Those synthesized CaO-SiO_2_ and MgO-SiO_2_ composites possessed high surface area, surface properties and thermal and chemical stability.

Alumina materials also possess high surface area, porosity, and thermal and mechanical stability. Therefore, researchers have recently used amine-grafted mesoporous silica and impregnated alumina as solid sorbents for CO_2_ capture [149]. Alumina-based materials for CO_2_ capture include basic Al_2_O_3_, amine-impregnated or amine-modified mesoporous Al_2_O_3_ and Al_2_O_3_–organosilica [149]. Gunathilake et al. [149] synthesized Al_2_O_3_–organosilica by introducing three different silica precursors such as tris [3-(trimethoxysilyl)propyl] isocyanurate (ICS), 1,4-bis(triethoxysilyl)benzene (BTEB), and bis(triethoxysilyl)ethane (BTEE)). This study used two alumina precursors, aluminum nitrate nanahydrate and aluminum isopropoxide, whereas grafting of amine groups was performed using 3-aminopropyltriethoxysilane (APTS). SiO_2_-Al_2_O_3_ showed the highest absorption capacity (Table 7), and the adsorption properties of the materials were dependent on the surface area of the sample, alumina precursor, and structure and functionality of the organosilica bridging group. Moreover, Choi et al. [152] used epoxy-functionalized PEI to synthesize CO_2_ sorbents. According to the reported data, epoxy-functionalized PEI exhibited a CO_2_ capacity of 2.2 mmol/g at 120 °C and 100% regeneration capability at similar temperatures. This can be attributed to the heat-resistant properties of epoxy butane, which enhanced the CO_2_ capture capacity and thermal stability of the silica-epoxy-PEI sorbent.

However, according to the reported data by Hu et al. [153], Li_4_SiO_4_ exhibited attractive prospects for CO_2_ capture. The main advantage of this material was the high CO_2_ sorption capacity (theoretical sorption capacity of 0.367 g CO_2_/g sorbent) and lower regeneration temperature (<750 °C) in comparison with other reported materials such as CaO, which requires a regeneration temperature of over 900 °C [153].

### 5.3. Sorbent Selectivity, Regeneration, and Stability in the Cyclic CO_2_ Adsorption–Desorption

During industrial applications, high adsorption capacity along with good regenerability of the sorbents in the cyclic adsorption–desorption process is vital [117]. The practical application of an adsorbent requires high sorption capacity, easy regeneration, stability in normal atmospheric conditions, and stable performance during cyclic use for long-term operation.

For instance, Ahmed et al. [93] reported a detailed study about the functionalization of mesoporous MCM-41 with different loadings of polyethylenimine (PEI). In this study, the selectivity measurement was conducted for CO_2_ over N_2_ and H_2_ and the adsorption capacities of N_2_ and H_2_ on 50 wt% PEI-Si-MCM-41 were 3.89 mg/g and 6.51 mg/g, respectively (see Table 8). Table 8 summarizes the gas selectivity values of previous studies performed for porous SiO_2_.

Wang et al. [154] prepared SBA-15 using silica-ethanol extraction and conventional high-temperature calcination template removal methods. Then, the silica was subjected to amine (3-aminopropyl) grafting and studied for its CO_2_ adsorption properties. This study aimed to increase the surface silanol density by grafting amine groups, increasing CO_2_ adsorption capacity and CO_2_/N_2_ selectivity. According to the reported data, CO_2_/N_2_ selectivity changed from 46 to 13 (see Table 8), and these results ensured that solvent extraction also enhanced CO_2_/N_2_ selectivity. Moreover, the authors performed a test to measure the stability of amine-SBA-15 (solvent extracted). According to the results, amine-SBA-15 (solvent extracted) was regenerated under a vacuum after each adsorption step.

In industrial applications of adsorbents, it is essential to remain stable during cyclic operations. This section summarizes the previous studies on sorbent regeneration and stability in cyclic CO_2_ adsorption–desorption by amine–silica composites, and the reported data are tabulated in Table 9. The regeneration of the amine-impregnated and grafted silica composites was mainly conducted by pressure and temperature swing adsorptions. Typically, the sorbent was regenerated at 50~120 °C in N_2_, He, or Ar flow. As depicted in Table 9, the amine-impregnated silica composites show a loss of CO_2_ capture capacity in the cyclic CO_2_ adsorption–desorption due to amine leaching from the silica surface and degradation [110]. Amine leaching is closely related to the amine types introduced and the operation temperature, while the degradation of amine is related to the operation temperature and gas atmosphere [109].

Guo et al. [128] conducted the adsorption/desorption cycles for hierarchically porous silica (HPS) grafted PEI at 75 °C. In this experiment, the modelled flue gas flow rate was maintained at 70 mL/min, and the CO_2_ partial pressure was held at 1 bar. According to the data, adsorption capacities are similar in eight adsorption/desorption cycles, showing that the aforementioned sorbents with good stability and regenerability.

Wang et al. [117] investigated the regenerability of the amine-modified MCM-41 (MCM-41-TEPA and MCM-41-AMP). The authors conducted fifteen cycles to verify the regenerability. According to the reported data, after fifteen cycles, the adsorption capacity decreased from 3.01 mmol/g to 2.88 mmol/g, and it was shown that both sorbents showed good regenerability. This may be due to the hydrogen-bonding interactions among TEPA, AMP and MCM-41, TEPA.

Kishor and Ghoshal [123] measured the stability of the pentaethylenehexamine (PEHA) impregnated KIT-6. The sorbent was aged for 6 months, and its adsorption performance was explored at 90–105 °C. The results showed that PEHA-impregnated KIT-6 had 4.0 and 4.3 mol CO_2_/kg sorption capacities at 90 and 105 °C at 1 bar even after 6 months. Moreover, the sorption performance of the adsorbent was tested for ten consecutive adsorption/desorption cycles. The sorption capacity of the sorbent decreased by less than 4% at 90–105 °C at 1 bar without any structural degradation. Moreover, the results exhibited that PEHA-impregnated KIT-6 had better sorption performance than those of earlier reported adsorbents, except for silica aerogel.

Liu and co-workers performed a regeneration test for zeolite-mesoporous silica-supported-amine hybrids sorbent [160]. Their data showed that, after 10 cycles, the adsorption capacity remained unchanged. Therefore, the sample performed a very stable cyclic adsorption–desorption performance. In contrast, López-Aranguren et al. [129] examined the regeneration of CO_2_ from branched PEI—mesoporous silica. In this study, CO_2_ adsorption–desorption cycles showed that the uptake measured in the first cycle was successfully maintained even after 20 cycles. Zhang et al. [174] examined the stability of the adsorbents based on linear PEI supported on silica. According to the reported data, the adsorbent maintained its adsorption capacity. Still, the adsorption capacity was reduced by approximately 5.6% when the temperature was increased to 100 °C, which was attributed to amine leaching. Furthermore, Subagyono et al. [162] found that the branched PEI-containing adsorbent decreased CO_2_ adsorption–desorption capacity during cycling, attributed to the by-product formation.

## 6. Technical Challenges and Future Trends

Financial, technical, and environmental concerns are the main barriers to CCS technologies. For instance, one major challenge with CCS is moving CO_2_ captured to remote storage sites using pipelines, as laying these pipelines is costly and associated with numerous environmental issues.

Several studies reported the requirements and a working definition for carbon dioxide capture (CCS). Advanced physical adsorbents must be developed with high CO_2_ selectivity and gas uptake. Stability (over 1000 cycles), CO_2_ affinity, scalability, reusability, resistance against surface erosion, and high energy requirement are the major concerns in CO_2_ capture technologies. The sorbent cost is the most significant part of an air capture system; however, it is difficult to estimate the price of a particular sorbent in lab-scale experiments. According to the reported data, the value of a kilogram of sorbent is equal to the net present value of the CO_2_ revenue collected during its lifetime. Therefore, a sorbent must possess constant stability and performance for its lifetime [178,179].

The other main challenges associated with sorbents are stability, kinetics, and sorbent capacity. However, many sorbents are thermodynamically strong enough to capture CO_2_ from ambient air and allow for easy regeneration. Despite the reported data, further studies on stability, kinetics and capacity still need to be improved in SiO_2_-based adsorbents. Another factor is sorbent loading and unloading cycles, which are essential for reducing costs. Moreover, adsorption kinetics is affected by binding energies, diffusion into porous materials, and the geometry of sorbent materials and many sorbents require longer sorption times. Therefore, improved kinetics can lower the cost. High adsorption capacity can reduce the cost of CO_2_ capture by reducing the amount of sorbent required. Physisorbents that selectively separate CO_2_ from gaseous mixtures formed a revolution in CCS since it requires less energy for recycling, with enhanced CO_2_ capacity.

Amine-based sorbents are widely used in CCS technologies. However, amine sorbent depends on the molecular weight of the sorbent and the pore sizes of the sorbent. To improve the capacity of moisture-swing sorbents, the ion exchange resins can be prepared with a higher charge density, and materials with different cation distances can be used under different humidity conditions. The potential of solid sorbents to remove CO_2_ from flue gas is enormous compared to conventional liquid amine processes in terms of regeneration energy and significant cost reduction. However, as discussed previously, solid sorbents have limitations and challenges to address before being deployed commercially in post-combustion CO_2_ capture.

There is limited literature available on CO_2_ capture using low-cost silica-based materials such as rice husks. These sources lead to the reduction in production costs. Nevertheless, novel silica-based materials such as lithium orthosilicate (Li_4_SiO_4_), silica nanotubes, silica nanospheres, silica-based composites, and silica aero gels give rise to high CO_2_ capture at elevated temperatures.

Moreover, most studies have used sol-gel and hydrothermal processes to synthesize silica-based sorbent. However, apart from the aforementioned methods, microwave treatment can also be used, which is cost-effective and timeserving. Moreover, different surfactants can prepare silica with varying pore sizes and morphologies. Another area for improvement with silica-based sorbent is the need for more literature on kinetic data at different adsorption temperatures, which are helpful in industrial implementations.

## 7. Summary

CO_2_ capture by porous SiO_2_ materials, their reaction mechanisms and synthesis processes were extensively discussed in this review. Chemical absorption of CO_2_ is more suitable than physical absorption owing to high adsorption capacity, relatively easy synthesis routes, and lower regeneration energy requirements. Among many chemisorbents, SiO_2_-based adsorbents, including amine-functionalized SiO_2_, possess higher CO_2_ selectivity and adsorption capacities, making them ideal candidates for CO_2_ capture. However, the performance of currently available amine-functionalized SiO_2_ needs to be further developed and improved in terms of stability, gas selectivity and resistivity to thermal degradation. Furthermore, the review highlighted major financial, technical, and environmental barriers and prospects associated with porous silica-based materials during the industrial scale-up process.

## Figures and Tables

**Figure 1 nanomaterials-13-02050-f001:**
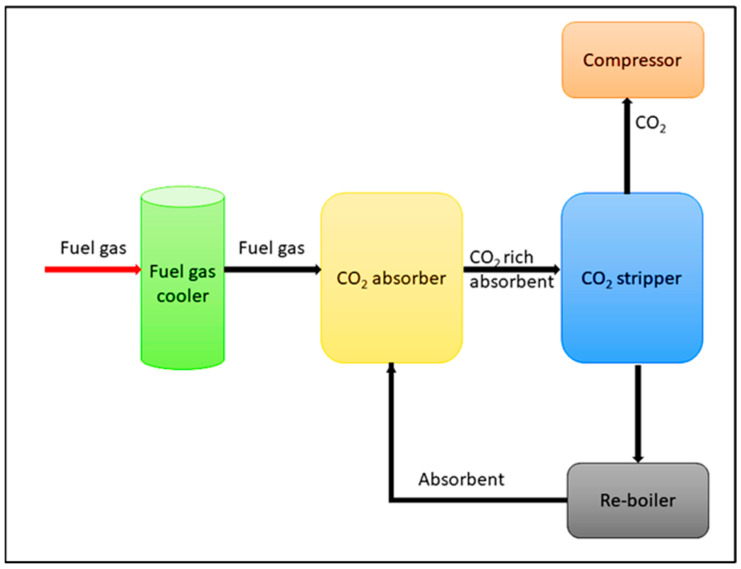
Schematic representation of post-combustion technology (Reprinted with permission from Osman et al. [1]).

**Figure 2 nanomaterials-13-02050-f002:**
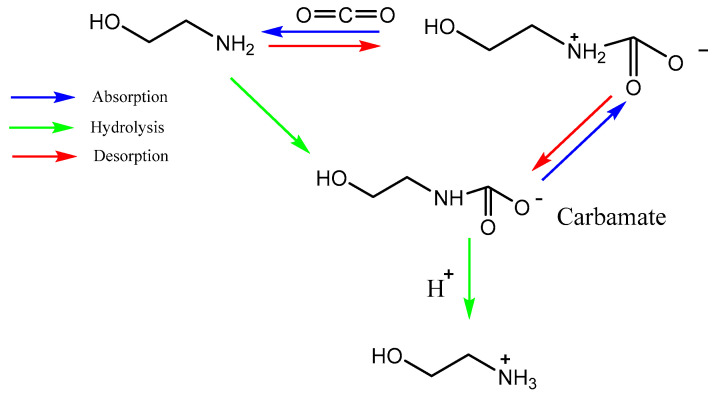
Reaction mechanism of CO_2_ capture into MEA solution (Reprinted with permission from Lv et al. [31]).

**Figure 3 nanomaterials-13-02050-f003:**
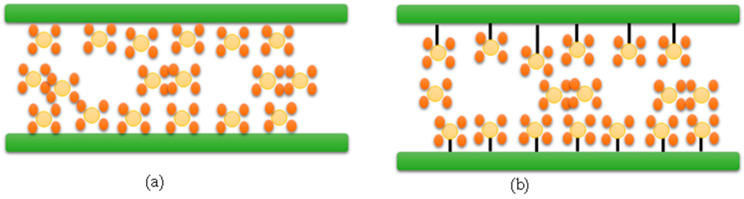
Schematic representation of (**a**) physisorption and (**b**) chemisorption.

**Figure 4 nanomaterials-13-02050-f004:**
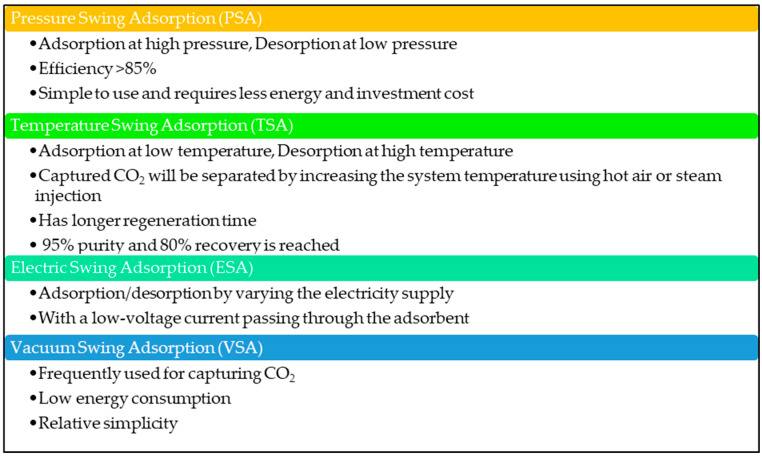
The different types of adsorption processes.

**Figure 6 nanomaterials-13-02050-f006:**
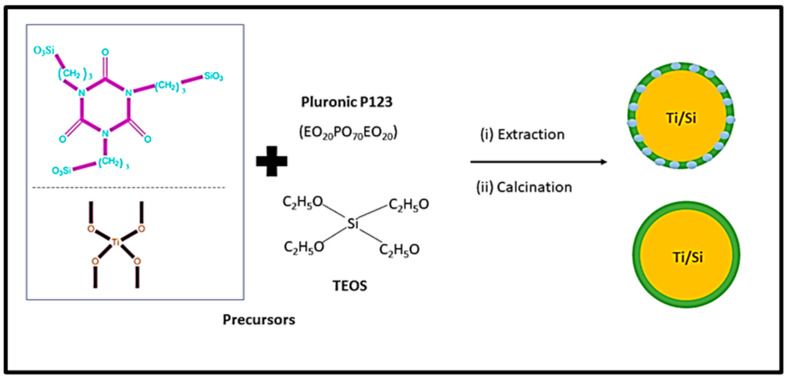
Mechanism for the synthesis of mesoporous silica using block copolymer (Re-printed with permission from Gunathilake et al. [88]).

**Figure 7 nanomaterials-13-02050-f007:**
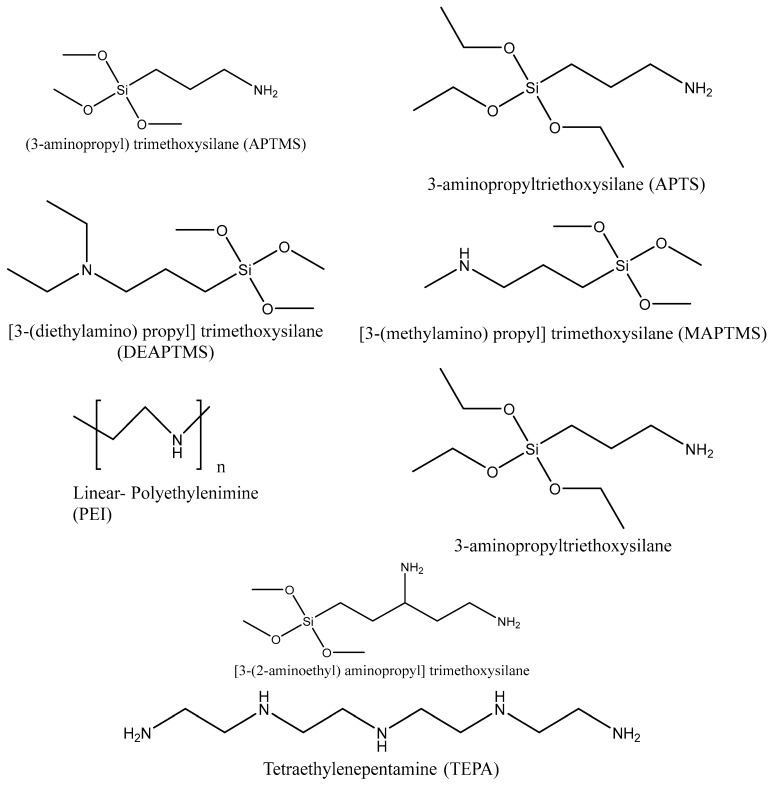
Amino silane- and polymer-containing amino groups used in the functionalization of mesoporous silicas.

**Figure 9 nanomaterials-13-02050-f009:**
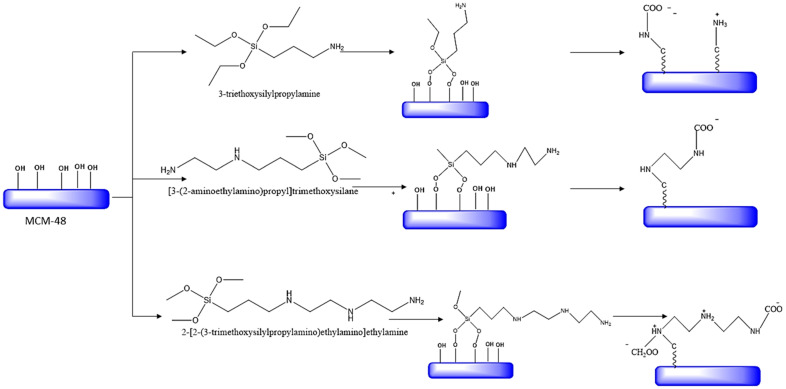
Schematic representation of the covalent bonding through the alkyl-silyl linkages and formation of carbamates (Reprinted with permission from Nigar et al. [99]).

**Table 1 nanomaterials-13-02050-t001:** Different approaches used in different countries in order to reduce the CO_2_ emissions [4].

Type of Approach	Details
Improve energy efficiency and promote energy conservation	This approach is mainly used in commercial and industrial buildings It shows mainly 10–20% of energy saving It shows extensive capital investment for installation
Increase in usage of low carbon or clean fuels such as natural gas, hydrogen or nuclear power; Substitution for Power generation	Natural gas emits 40–50% less CO_2_ than coal Main advantages of this method are high efficiency and cleaner exhaust gas Main disadvantage is the high cost
Deploy renewable energy	The renewable energy sources include solar, wind, hydropower, geothermal, oceanic energy and bioenergy This method emits low green house and toxic gases The main limitation is high cost and geographic distribution of the available resources
CO_2_ capture and storage	This method is applicable for large CO_2_ point emission sources It can reduce vast amount of CO_2_ with capture efficiency of 48%

**Table 2 nanomaterials-13-02050-t002:** Comparison of different post-combustion capture technologies for CO_2_ capture.

Technology	Types	Examples	Efficiency (%)	Advantages	Disadvantages	Ref.
**Absorption**	Chemical	Amines Caustics	>90	Ability to regenerate Established method Very flexible Reacts rapidly High absorption capacities	High energy requirement for regeneration Environmental problems High boiling point Equipment corrosion	[21,22]
	Physical	Selexol Rectisol fluorinated solvents				
**Adsorption**	Chemical	Metal Oxides Si based materials	>85	Recyclable Cost effective High stability Adjustable catalytic site and pore sizes Low energy consumption Suitable for separating CO_2_ from dilute streams	High energy cost Limited to process feed rates Loss of material and pressure drop Decreased catalytic efficiency Low adsorption capacities	[6,21]
	Physical	Carbons Zeolites Si based materials				
**Membrane-based technologies**	Organic Cellulose derivatives Polyamides	Polyphenyleneoxide, Polydimethylsiloxane	>80	Simple device Easy production process and process flow scheme Low energy consumption No phase changes Capable of maintaining the membrane structure	Requires a high-cost module and support materials Not suitable for large volumes of emission gases Reduced selectivity and separation Pressure drops across the membrane Less durability	[6,21]
	Inorganic	Metallic Ceramics				
**Cryogenic distillation**				Low capital investment High reliability Recovery with high purity of CO_2_ Liquid CO_2_ production Not requiring solvents or other components Easily scalable to industrial-scale applications	High energy consumption	[6,21,23]

**Table 4 nanomaterials-13-02050-t004:** Advantage and disadvantages of non-carbonaceous adsorbents.

Material Types	Examples	Advantages	Disadvantages
**Pours silica** **materials**	M41S SBA-n AMS	High specific surface area, Pore volume, and good thermal and mechanical properties	High molecular diffusion resistance Decreased adsorption capacity at high temperature [42]
**Zeolites**	NaY 13X	Low production cost Large micropores/mesopores Medium CO_2_ adsorption capacity at room temperature	Low CO_2_ adsorption capacity Moisture-sensitivity High energy consumption [6,43]
**Metal organic** **frameworks (MOFs)**	M-MOF-74 IRMOF-6 USO-2-Ni Zn_4_O (BDC)_3_ (MOF-5) USO-1-Al (MIL-53)	Large specific surface area Ease of controlling pore sizes High selectivity of CO_2_	Low CO_2_ adsorption capacity at the partial pressure High production cost Complicated synthesis process Moisture-sensitivity Unstable at high temperature [6]
**Alkali-based dry** **adsorbents**		Possible adsorption and desorption at a low temperature and wet conditions	Low adsorption capability (3–11 wt%) High-temperature reactions Requires high temperatures during desorption Complicated operation [6]
**Metal oxides-based** **adsorbents**	CaO, MgO	Dry chemical adsorbents Adsorption/desorption at medium to high temperatures	High energy consumption High cost for regeneration Complicated process [6]

**Table 5 nanomaterials-13-02050-t005:** Comparison between chemisorption and physisorption.

	Chemisorption	Physisorption
**Description**	Chemical reaction occurs between the solid sorbents and CO_2_	Depends on the physical properties of CO_2_ and the ability to engage in noncovalent interactions with the solid sorbent
**Chemical Bonding**	Covalent Bonding-Occur between functional groups and CO_2_ in the surface	Week Vander-walls forces-London and Dispersion forces, Occur inside pore walls
**Advantages**	High selectivity	Low recycling energy requirements High working capacity High selectivity even in wet environments Fast
**Disadvantages**	High energy required for recycling and the breakage of the chemical bonds Slow reactivity	Poor selectivity in binary or mixed gas applications
**References**	[55,56]	[25,57,58]

**Table 8 nanomaterials-13-02050-t008:** Summary of gas selectivity values of previous studies performed for porous SiO_2_.

Porous SiO_2_ Material	Gas Mixture	Selectivity Value	Pressure (Bar)	Temperature (°C)	Reference
**PEI-MCM-41**	CO_2_, N_2_ and H_2_	25.56	1	100	[93]
**SBA-15**	CO_2_/N_2_	123	1	25	[154]
**SBA-15 (calcination)**	CO_2_/N_2_	55	1	25	[154]
**Mesoporous chitosan−SiO_2_ nanoparticles**	-	15.46	1	25	[155]
**hydrophobic microporous high-silica zeolites**	CH_4_:N_2_ = 50%:50%	36.5	1	25	[156]
**Hollow silica spherical particles (HSSP)**	CO_2_/N_2_	8.5	4	25	[157]
**microporous silica xerogel**	CO_2_/CH_4_	60	6	25	[158]
**Silica based xerogels**	C_2_H_4_/C_2_H_6_	20	6	25	[158]

**Table 9 nanomaterials-13-02050-t009:** Summary of stability of silica-based adsorbent studied in past performance capacity.

Synthesis Method	Type of Silica-Based Sorbent	Amine Type	Regeneration Condition		Stability Performance		References
			Temperature (°C)	Types of Gas Flow	No. of Cycles (Cyclic Runs)	Capacity Loss (%)	
**Impregnated**	MCM-41	PEHA	100	N_2_	15	Less than 1	[159]
	MCM-41	TEPA + AMP	100	N_2_ for 60 min	15	4.32	[117]
	SBA-15	PEI-linear	100	Ar	12	13.5	[160]
	SBA-15	Acrylonitrile-modified TEPA	100	N_2_	12	1.1	[161]
	HMS	PEI-linear	75	N_2_ for 100 min	4	1.6	[110]
	MCF	PEI-branched	115	Ar for 20 min	10	32	[162]
	MCF	PEI	100	H_2_	10	5	[163]
	MCF	Guanidinylated poly(allylamine)	120	He	5	17	[52]
	Fumed silica	PEI-linear	55	N_2_ for 15 min	180	Stable	[164]
	MCM-41	TEPA	100	N_2_	10	3.43	[165]
	Silica fume	Diisopropanolamine	50	N_2_	10	7	[166]
	Nano-SiO_2_	PEI-branched	120	N_2_	30	10.5	[167]
	Nano-SiO_2_	PEI-branched	120	N_2_	30	19.4	[168]
	Mesoporous-SiO_2_	APTS	120	Air for 30 min	11	4.3	[169]
	Porous SiO_2_	PEI	100	N_2_ for 30 min	20	5	[170]
	Silica aerogel	TEPA	75	Ar for 20 min	10	3.9	[171]
	Porous SiO_2_	TEPA	75	He for 20 min	10	2	[172]
	SNT	PEI	110	N_2_ for 40 min	10	3.3	[132]
	KCC-1-SiO_2_	TEPA	110	N_2_	21	1.2	[173]
	Mesoporous multilamellar SiO_2_	PEI	110	N_2_	10	3.7	[174]
	Silica aerogel	TEPA	80	Ar for 30 min	100	12	[173]
	Mesoporous SiO_2_	DEA	90	N_2_	10	12	[169]
**Grafting**	SBA-15	AP	90	Vacuum	10	1	[175]
	SBA-15	DEAPTMS	120	N_2_ for 10 min	100	7.2	[176]
	MCM-48	2-[2-(3-trimethoxysilyl propylamino) ethylamino] ethylamine	-	N_2_	20	Stable	[98]
	KIT-6	APTES	120	He	10	Stable	[97]
	MCF	TRI	150	N_2_ for 30 min	5	1.9	[177]
	HMS	APTS	110	N_2_ for 180 min	3	Less than 1	[178]
	MCM-41	APTS	105	N_2_ for 90 min	10	Stable	[115]

## Data Availability

Data sharing not applicable.

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
