# Peer review of "Carbon Capture Using Porous Silica Materials"

_nanomaterials, 2023, doi:10.3390/nano13142050_

Round 1

Reviewer 1 Report

The author has compiled a review article by gathering relevant literature on carbon dioxide capture techniques and capture methods from the past decade. The article also provides a detailed comparison of the advantages and disadvantages of amine-based chemical absorbents and inorganic porous adsorbents, which offers significant reference value for researchers in this field.  The article specifically focuses on the application of porous silica materials in CO2 capture. Due to the ability of silica to undergo surface modification using various silane coupling agents with different functional groups, it enhances the adsorption performance for CO2. Based on the findings from previous literature studies, amine-functionalized SiO2 possesses higher CO2 selectivity over other gases and high CO2 adsorption capacities which make them ideal candidates for CO2 capture. It provides high reference value for future applications.  Hence, I can recommend to accept this paper.

Reviewer 2 Report

After getting familiar with the reading-matter of reviewed article, I state the following:
1. The article presents a comprehensive study of currently existing CO2 capture techniques with a particular discussion of the adsorption of gaseous CO2 on various types of amine-functionalized porous silica materials.
2. The title of the article fully reflects the content of the article.
3. The article is very valuable, well written and understandable to a wide range of readers.
4. Due to the large number of acronyms in the text, their list should be included in the text (at the beginning or at the end of the article), it will make it easier for readers to read.
5. Line 23 is "porous sloid sorbents", it should be "porous solid sorbents".
6. Line 827 - These are not conclusions, rather summary.
7. The article is based on numerous literature sources.
8. The article meets the substantive requirements of the journal Nanomaterials.
